# miR-335 Restrains the Aggressive Phenotypes of Ovarian Cancer Cells by Inhibiting COL11A1

**DOI:** 10.3390/cancers13246257

**Published:** 2021-12-13

**Authors:** Yi-Hui Wu, Yu-Fang Huang, Tzu-Hao Chang, Pei-Ying Wu, Tsung-Ying Hsieh, Sheng-Yen Hsiao, Soon-Cen Huang, Cheng-Yang Chou

**Affiliations:** 1Department of Medical Research, Chi Mei Medical Center, Liouying, Tainan 736, Taiwan; a80869@mail.chimei.org.tw; 2Department of Nursing, Min-Hwei Junior College of Health Care Management, Tainan 736, Taiwan; 3Department of Obstetrics and Gynecology, National Cheng Kung University Hospital, College of Medicine, National Cheng Kung University, Tainan 704, Taiwan; yufangh@mail.ncku.edu.tw (Y.-F.H.); anna1002ster@gmail.com (P.-Y.W.); 4Graduate Institute of Biomedical Informatics, Taipei Medical University, Taipei 110, Taiwan; kevinchang@tmu.edu.tw; 5Department of Obstetrics and Gynecology, Chi Mei Medical Center, Liouying, Tainan 736, Taiwan; eric99.hsieh@msa.hinet.net (T.-Y.H.); d930817@mail.chimei.org.tw (S.-C.H.); 6Department of Hematology and Oncology, Chi Mei Medical Center, Liouying, Tainan 736, Taiwan; seedvirt@gmail.com

**Keywords:** epithelial ovarian carcinoma, collagen type XI alpha 1, miR-335, clinical oncology, COL11A1, EOC cells, miRNA, miR-509

## Abstract

**Simple Summary:**

High collagen type XI alpha 1 (COL11A1) levels are associated with tumor progression, chemoresistance, and poor patient survival in several cancer types. We examined the role of miRNAs in regulating COL11A1 expression. This study highlighted the importance of miR-335 in downregulating COL11A1-mediated ovarian tumor progression, chemoresistance, and poor survival and suggested its potential application as a therapeutic target.

**Abstract:**

High collagen type XI alpha 1 (COL11A1) levels are associated with tumor progression, chemoresistance, and poor patient survival in several cancer types. MicroRNAs (miRNAs) are dysregulated in multiple cancers, including epithelial ovarian carcinoma (EOC); however, the regulation of COL11A1 by miRNAs in EOC remains unclear. We examined the role of miRNAs in regulating COL11A1 expression. We identified miR-509 and miR-335 as the candidate miRNAs through an online database search. EOC cell treatment with miR-335 mimics abrogated COL11A1 expression and suppressed cell proliferation and invasion, besides increasing the sensitivity of EOC cells to cisplatin. Conversely, treatment with miR-335 inhibitors prompted cell growth/invasiveness and chemoresistance of EOC cells. miR-335 inhibited COL11A1 transcription, thus reducing the invasiveness and chemoresistance of EOC cells via the Ets-1/MMP3 and Akt/c/EBPβ/PDK1 axes, respectively. Furthermore, it did not directly regulate PDK1 but increased PDK1 ubiquitination and degradation through COL11A1 inhibition. In vivo findings highlighted significantly decreased miR-335 mRNA expressions in EOC samples. Furthermore, patients with low miR335 levels were susceptible to advanced-stage cancer, poor response to chemotherapy, and early relapse. This study highlighted the importance of miR-335 in downregulating COL11A1-mediated ovarian tumor progression, chemoresistance, and poor survival and suggested its potential application as a therapeutic target.

## 1. Introduction

MicroRNAs (miRNAs) are approximately 22-nucleotide-long noncoding RNAs, which are highly conserved among a wide range of species and are generally involved in post-transcriptional gene regulation. Often, miRNAs negatively regulate gene expression by binding to the 3′-untranslated region (UTR) of target mRNAs. They do not require perfectly complementary target sites and recognize short sites, complementary to their 5′-seed region (nucleotides 2–8 of the miRNAs); thus, one miRNA can regulate numerous mRNAs, and multiple miRNAs can regulate an individual mRNA [1]. These miRNAs supposedly regulate approximately 60% of all human genes and are involved in processes such as development, differentiation, metabolism, proliferation, cell cycle, inflammation, and the regulation of the immune system [2,3,4,5]. Currently, several miRNAs, either upregulated or downregulated, are implicated in the initiation, progression, and dissemination of various cancer types [6,7]. Multiple studies have reported the dysregulation of miRNAs in ovarian cancer and suggest their pathobiological importance, such as the regulation of cell proliferation, survival, and stemness, by either the targeted inhibition of tumor suppressor genes or the activation of oncogenes [8,9,10,11,12,13].

Collagen type XI alpha 1 (COL11A1) belongs to the collagen family, which is the major component of the interstitial extracellular matrix. COL11A1 expression is upregulated in several cancer types, including ovarian cancer, breast cancer, pancreatic cancer, non-small-cell lung cancer, and colorectal cancer. High COL11A1 levels are associated with tumor aggressiveness, chemoresistance, and poor survival in various solid tumor types [14,15]. In addition, COL11A1 can serve as a specific marker for cancer-associated fibroblasts (CAFs) [16,17,18]. COL11A1 overexpression promotes the progression of epithelial ovarian cancer (EOC) progression and induces chemoresistance [19,20,21,22,23,24]. Moreover, it mediates CAF activation, crosstalk between cancer cells, and constituents in the tumor microenvironment to regulate cancer cell phenotypes [16].

MicroRNAs may influence COL11A1 expression. miR-139-5p [25] and miRNA let-7b [26] regulate COL11A1 in breast cancer cells. Moreover, miR-29 is likely involved in colon cancer progression by downregulating COL11A1 [27]. However, the miRNA-mediated regulation of COL11A1 in EOC cells is still unclear.

In this study, we ascertained that miR-335 acts as a novel tumor suppressor in ovarian cancer. The tumor-suppressive function of miR-335 is mediated by the suppression of COL11A1 expression, thereby reducing the invasive ability and chemoresistance of EOC cells via the Ets-1/MMP3 and Akt/c/EBPβ/PDK1 axes, respectively. We further revealed the clinical relevance of miR-335 in ovarian cancer.

## 2. Materials and Methods

### 2.1. Cells and Media

The human ovarian cancer ES-2 cell line was purchased from the Bioresource Collection and Research Center of the Food Industry Research and Development Institute (Hsinchu, Taiwan). The OVCAR-3 and OVCAR-8 cell lines were purchased through the National Cancer Institute DTP tumor repository program. A2780 and A2780CP70 cell lines were provided by Dr. Hsu Keng-Fu (Department of Obstetrics and Gynecology, National Cheng Kung University Hospital, College of Medicine, National Cheng Kung University, Tainan, Taiwan). We grew the OVCAR-3, OVCAR-8, A2780, and A2780CP70 cells in Roswell Park Memorial Institute-1640 medium, supplemented with 10% fetal bovine serum. ES-2 cells were grown in Mycos 5A medium, supplemented with 10% fetal bovine serum, at 37 °C in a 5% CO_2_ atmosphere. These cells were cultured and stored according to the supplier’s instructions and used between 5 and 20 passages. Once thawed, the cell lines were routinely authenticated approximately every 6 months. They were last tested in March 2021, through cell morphology monitoring, growth curve analysis, species verification by isoenzymology, and karyotyping identity verification, using short tandem repeat-profiling analysis and contamination checks. MG132 was obtained from Sigma-Aldrich (St. Louis, MO, USA).

### 2.2. Cell Transfection

We purchased miR-335 mimics (MC10063), miR-509-3p mimics (MC12984), miRNA mimic negative control (4464058), miR-335 inhibitor (MH10063), miR-509 inhibitor (MH12984), and inhibitor negative control (4464076) from Ambion (Foster City, CA, USA). COL11A1 cDNA (BC117697 GE Healthcare, Chicago, IL, USA) was cloned into the pCMV6-AC-GFP vector (PS100010 OriGene) and checked by sequencing. miR-335 mimics combination with pCMV6-AC-GFP vector carrying COL11A1, which was transfected into the A2780CP70 or OVCAR-8 cell using Lipofectamine 3000 (Thermo Fisher Scientific, Waltham, MA, USA).

### 2.3. miRNA Isolation

All ovarian cancer specimens and non-cancerous controls were de-identified and collected, with the approval of institutional review boards. We extracted total RNA from 100 mg fresh tissue samples with at least 70% tumor cellularity using the miRNeasy kit (Qiagen, Hilden, Germany). The extracted RNAs were assessed for quality and quantity using an Agilent 2100 Bioanalyzer and NanoDrop 1000 spectrophotometer (Thermo Scientific).

### 2.4. miR-335 and miR-509 Measurement

We performed a quantitative real-time polymerase chain reaction (qPCR), according to the manufacturer’s instructions (Qiagen), to profile the miRNA distribution in the collected samples. In brief, 100 ng total RNA was collected and pooled from the samples, and the cDNA was produced using the miScript Reverse Transcription kit (Qiagen). We used the Matrix Hydra eDrop (Thermo Scientific) to mix the cDNA sample and qPCR master reagent (Human miScript Assay 384 set v10.1 [Qiagen]) to reduce pipetting error. Wells with multiple melting-temperature values were excluded from further analysis. Data were normalized with the global mean instead of specific miRNA or noncoding RNA signals. We calculated miRNA expression by utilizing the comparative C_t._ method. Statistical analysis was performed using the Student *t*-test; *p*-values < 0.05 were considered statistically significant.

### 2.5. Quantitative Reverse Transcriptase PCR

Total RNA (5 μg) was used as the template in cDNA synthesis reactions, with random primers using Superscript III reverse transcriptase (Promega, Madison, WI, USA), and the resultant cDNAs were used (at a 1:20 dilution) to detect the level of endogenous and *COL11A1* mRNA expression by quantitative PCR. We used the StepOnePlus Real-Time PCR System (Thermo Fisher Scientific, Waltham, MA, USA) to quantitatively analyze mRNA expression. The primers and TaqMan probes used for the analyses were designed using the manufacturer’s software, Primer Express. We used the following primers: *COL11A1* (HS01097664) and *GAPDH* (HS99999905). No reverse transcription control reactions were performed, using 3 μg of total RNA from each sample as a template. All quantitative analyses were performed in duplicate. The relative expression levels of the target gene, normalized to *GAPDH* expression, were calculated as ΔC_t_ = C_t_ (target) − C_t_ (*GAPDH*). Relative fold changes in gene expression were calculated using the comparative 2^−ΔΔCT^ method [28].

### 2.6. Western Blot Analysis, Antibodies, and Reagents

Following protein extraction, equal amounts were separated by 8–15% sodium dodecyl sulphate–polyacrylamide gel electrophoresis [19]. Antibodies against COL11A1 (GTX55142) were obtained from GeneTex (Irvine, CA, USA). Antibodies against β-actin (sc-47778) and Ets-1 (sc-111) were purchased from Santa Cruz Biotechnology (Dallas, TX, USA), whereas those against Akt (9272), phospho-Akt (p-Akt, 9271), PDK1 (3062), mouse IgG (7076), and rabbit IgG (7074) were obtained from Cell Signaling Technology (Danvers, MA, USA). Cisplatin (Fresenius Kabi Oncology, Ltd., Homburg, Germany) was provided by the Cancer Center of National Cheng Kung University Hospital. The full western blots Figures are shown in Appendix A.

### 2.7. Luciferase Reporter Analysis

COL11A1 3′-UTR or PDK1 3′-UTR fragments with wild-type miR-335 binding sites (Wt) or mutated binding sites (Mut) were inserted into the pGL4 vector (Promega). The COL11A1 3′-UTR or PDK1 3′-UTR PCR product was cloned into the *Sac*I and *EcoR*V sites of the pGL4 vector. Primers of COL11A1 3′-UTR with the following sequences were used: region 1 forward 5′-GTAATTACGACGACACACTTCT-3′ and reverse 5′-AGAAGTGTGTCGTCGTAATTAC-3′; region 2 forward 5′-TATGCAGCATCTTGCTACATTCA-3′ and reverse 5′-TGAATGTAGCAAGATGCTGCATA-3′; region 3 forward 5′-AGATTTTCAAAACTATTAACACCTT-3′ and reverse 5′-AAGGTGTTAATAGTTTTGAAAATCT-3′. Primers of PDK1 3′-UTR with the following sequences were used: forward 5′-GAGCTCGTGGTTCACAAGAGCCCAGC-3′ and reverse 5′-GATATCAATTTGGGTGGTCTGGACTT-3′. The resultant construct was confirmed by DNA sequencing. Site-directed mutagenesis was used to generate COL11A1 3′-UTR or PDK1 3′-UTR constructs, containing miR-335 mutant-binding sites using complementary oligonucleotides (Appendix A). The vector combination with miR-335 mimics was transfected into A2780CP70 or OVCAR-8 cells. We performed luciferase assays 48 h post-transfection using the dual-luciferase reporter assay system (Promega, Madison, WI, USA). Normalized luciferase activity is reported as the ratio of luciferase to β-galactosidase activity. Firefly luciferase and *Renilla* luciferase activities were measured as described previously [29].

### 2.8. Transwell Invasion Assay

We examined cell invasion in Transwell cell culture chambers using polycarbonate membranes with 8 μm pores (Costar, Cambridge, MA, USA). The upper chamber of the Transwell membranes was coated with rat collagen I (60 µg/Transwell) and filled with A2780 or A2780CP70 cells (5 × 10^4^). The lower chamber contained 0.6 mL of the medium containing fibronectin as a chemoattractant. The cells were allowed to invade for 24 h, at 37 °C, under 5% CO_2_. Non-migrated cells in the upper chamber were removed with a cotton swab, and the filters were fixed in 95% ethanol and stained with 0.005% crystal violet for 1 h. We used a phase-contrast microscopy (Olympus, Lake Success, NY, USA) to count the cells that had migrated to the lower surface. Ten contiguous fields were examined to obtain a representative number of cells. Each condition was assayed in triplicate. The invasive capacity of a treated sample was normalized to that of the corresponding control. One-sample unpaired Student’s *t*-test was used to test the differences of normalized invasive capacities of three independent experiments with the hypothetical value (set to 1).

### 2.9. Casein Zymography Analysis

To measure MMP3 activity, the conditioned medium from treated cells was concentrated approximately 20-fold using Centricon-10 spin concentrators (Millipore, Billerica, MA, USA). The samples were quantified by Bradford analysis, and equal amounts of protein were mixed with Laemmli sample buffer without reducing agents, incubated for 15 min at 37 °C, and separated on precast gradient SDS-polyacrylamide slab gels containing 1 mg/mL casein (Sigma, MO, USA). Following electrophoresis, the gels were placed in 2.5% Triton X-100 for 30 min and incubated at 37 °C in 50 mM Tris–HCl with pH 7.4, containing 5 mM CaCl_2_ for 18 h. MMP3 activity was visualized by Coomassie blue staining.

### 2.10. 3-(4,5-Dimethylthiazol-2-yl)-2,5-diphenyltetrazolium Bromide Cytotoxicity Assay

The analysis was conducted as previously reported [20].

### 2.11. Study Population

The research protocol was approved by the National Cheng Kung University Hospital Institutional Review Board (No. B-ER-107-414) and Institutional Review Board of Chi Mei Medical Center (10808-L03). We included 137 patients with EOC and 23 non-cancerous patients, between 2010 and 2020, in determining miR-335 and *COL11A1* expression levels in intraoperatively collected primary ovarian cancerous specimens and benign ovarian lesions, respectively. The staging was performed according to the International Federation of Gynecology and Obstetrics criteria. Cancer progression was defined according to the objective response evaluation criteria in solid tumors 1.1 or the Gynecologic Cancer Intergroup definition for CA 125 progression. Patients with EOC were followed up after treatment, and the date of the latest retrieved record was 15 August 2021. We calculated both progression-free survival (PFS) and overall survival (OS) from the diagnosis. Medical records and pathology slides were reviewed for the demographic data, clinical characteristics, progression-free interval (PFI), and treatment outcomes. The PFI was measured from the date of completion of front-line, platinum-based chemotherapy to disease progression or last contact. Patients with EOC and a PFI <6 months or ≥6 months were categorized as “resistant” or “sensitive” to platinum-based chemotherapy, respectively.

### 2.12. Statistical Analyses

Continuous variables are expressed as mean ± standard deviation or median ± interquartile range (range), following normality testing. On the contrary, categorical variables are presented as frequencies and percentages. We analyzed data using SPSS version 21.0 (IBM Corp., Armonk, NY, USA). Between-group differences of continuous variables were analyzed using the Mann–Whitney U test. We performed the Pearson’s chi-squared test and Fisher’s exact method to compare frequency distributions between categorical variables. The receiver operating characteristic curve-determined cut-off value of miR-335 and COL11A1 was optimized for the diagnostic sensitivity and specificity to predict cancer progression or death. Survival was estimated using the Kaplan–Meier method and compared with the log-rank test; *p* < 0.05 (two-sided) was considered statistically significant.

## 3. Results

### 3.1. Investigating COL11A1 miRNA Target Interactions with the Online Database

To investigate regulatory miRNAs of the COL11A1 gene, we used the online resource platform miRWalk [30] for target mining. We searched for the COL11A1 gene with a binding probability >0.5 in the miRWalk database and identified 596, 1238, and 8874 miRNA target interactions (MTIs) of 3′UTR, 5′UTR, and CDS of COL11A1 gene, respectively (Appendix A). Considering that each miRNA could have more than one binding site of one gene, we identified fewer miRNAs.

Then, we downloaded TCGA ovarian cancer RNA sequencing RPKM data and miRNA sequencing RPM data of 299 and 461 samples from Broad GDAC firehose, respectively. For the integrated analysis, we selected 292 samples with both RNA and miRNA expression data. We defined a threshold of miRNA expression as an average RPM ≥ 1, and only 425 miRNAs were left. To investigate the potential relationship between the COL11A1 gene and miRNAs, we calculated the Spearman correlation of miRNA and COL11A1 mRNA expression data (Appendix A).

After calculating the correlation between miRNAs and COL11A1 mRNA expression, we selected 36 miRNAs with Spearman correlation <−0.195. These 36 miRNAs were combined with the list of miRNAs selected from the miRWalk database. Subsequently, we identified three common miRNAs (hsa-miR-509-3p, hsa-miR30e-5p, and hsa-miR-877-5p) with predicted miRNA target interactions (MTIs) on the 3′UTR of COL11A1 gene (Appendix A). The hsa-miR-509-3p displayed the top one miRNA correlation with COL11A1; therefore, it was selected for further experimental analysis.

Apart from the aforementioned analysis, we used miRTarBase [31], the experimentally-validated miRNA target interactions database, to search for the validated miRNA target interaction of the COL11A1 gene. One miRNA (hsa-miR-335-5p) was recorded in miRTarBase with MTIs validated by microarray. Eventually, hsa-miR-335-5p was selected for further experimental analysis.

### 3.2. miR-335/miR-509 Expression in EOC Cells and Their Correlation with COL11A1 Expression

We examined if miR-509 and miR-335 could regulate COL11A1 transcriptions. The mRNA expressions of COL11A1 were analyzed by real-time reverse transcription (RT)-PCR in either overexpressing or inhibiting miRNAs in ovarian cancer cells. COL11A1 mRNA expressions were negatively correlated with miR-335 levels in a panel of EOC cell lines. Compared with its cisplatin-resistant counterpart-A2780CP70 cells, the chemosensitive low COL11A1-expressing A2780 cells exhibited lower COL11A1 and higher miR-335 levels. Similarly, high COL11A1-expressing OVCAR-8 cells exhibited lower miR-335 levels than OVCAR-3 cells. The miR-509 levels in EOC cell lines were supposedly negatively correlated with COL11A1 levels (Figure 1A). However, COL11A1 expression increased following EOC cell treatment with miR-335 inhibitor (A2780 and OVCAR-3), whereas COL11A1 decreased after miR-335 mimics treatment (A2780CP70 and OVCAR-8, Figure 1B). In contrast, the COL11A1 expression did not change following the miR-509 inhibitor or miR-509 mimic treatment (Figure 1C). These results showed that COL11A1 could not be regulated by miR-509.

Subsequently, we used miRTarBase to search for the validated miR-335 target interaction of the COL11A1 gene. The left panel denoted three predicted miR-335 binding regions on the COL11A1 3′-UTR by miRanda algorithm (Figure 1D). To determine the binding region involved, we constructed *COL11A1* promoters with mutations in these binding sites using site-directed mutagenesis. The luciferase reporter analysis revealed that the co-transfection of miR-335 and COL11A1-Wt, COL11A1-Mut1 (region 377–401), and COL11A1-Mut2 (region 715–739) degraded luciferase activities in A2780CP70 cells. In contrast, luciferase activities in A2780CP70 cells co-transfected with COL11A1-Mut3 (region 1005–1032) and miR-335 were not significantly reduced (Figure 1D, right panel). In other words, the miR-335 binding region (1005–1032) on the COL11A1 promoter was the major determinant of the COL11A1 gene negatively regulated by miR-335 in EOC cells.

### 3.3. Invasive Phenotypes of EOC Cells Are Regulated by miR-335

To ascertain the impact of miR-335 on the aggressive traits of OC cells, A2780CP70 and OVCAR-8 cells were transfected with miR-335 mimics to raise the expression of miR-335. Moreover, the miR-335 inhibitor was transfected into A2780 and OVCAR-3 cells to decrease miR-335 expression (Figure 2A). The cell viabilities of miR-335 mimic transfected cells were strikingly restrained (upper panel, Figure 2B), whereas the cell viabilities in miR-335 inhibitor transfected cells were increased (lower panel, Figure 2B). We further explored whether COL11A1 was critical for the impacts of miR-335. Cells were transfected with pCMV6-AC-GFP vector carrying COL11A1 to rescue the endogenous expression of COL11A1 that was inhibited by miR-335 (upper panel, Figure 2C). The result of MTT experiments implied that re-expression of COL11A1 abolished the inhibiting influences of miR-335 on OC cell growth (lower panel, Figure 2C).

Transwell invasion assays revealed less invaded cell number in the miR-335 mimic transfected cells than those in controls (upper panel, Figure 2C). Moreover, the invaded cell number increased in miR-335 inhibitor transfected cells (lower panel, Figure 2C). Previously, we indicated that COL11A1 promotes cell aggressiveness via the TGF-β1/Ets-1/MMP3 axis [19]. Ets-1/MMP3 expressions and MMP3 activity, measured by casein zymography, decreased by miR-335 mimic and increased by a miR-335 inhibitor (Figure 2D). Therefore, miR-335 inhibited the aggressive phenotypes in EOC cells.

COL11A1 confers chemoresistance on EOC cells, through the activation of the Akt/c/EBPβ pathway and PDK1 stabilization (20). We examined the miR-335 regulation of COL11A1-mediated chemoresistance in EOC cells. The cell sensitivity to cisplatin reduced in the miR-335 mimic transfected cells (IC_50_ value from 22.61 μM to 5.27 μM, *p* < 0.01) and increased in the miR-335 inhibitor transfected cells (IC_50_ value from 6.09 μM to 28.52 μM, *p* < 0.01, Figure 3A) in a dose-dependent manner. The expressions of p-Akt, COL11A1, and PDK1 were attenuated by miR-335 mimic treatment (Figure 3B). To determine if miR-335 can directly regulate PDK1, we searched for the candidate target genes using TargetScanHuman 7.2, the online miRNA target database (www.targetscan.org, last accessed on 25 March 2021). The predicted miR-335 binding position on PDK1 3′UTR is observed in the upper panel of Figure 3C. Luciferase activities in A2780CP70 cells, co-transfected with miR-335 and PDK1-Wt or PDK1-Mut, did not decrease (Figure 3C, lower panel). Further experiments revealed that the pattern of PDK1 ubiquitination, following MG132 treatment, was more extensive in A2780CP70 cells treated with miR-335 mimic than in those without miR-335 mimic treatment (Figure 3D). In other words, miR-335 did not directly regulate PDK1 but increased PDK1 protein ubiquitination and degradation through COL11A1 inhibition.

### 3.4. miR-335 Expression and Its Clinical Implications

miR-335 expression levels in EOC specimens were significantly lower than those in non-cancerous specimens (*p* = 0.002). We observed a correlation in miR-335 and COL11A1 levels between tissue samples of 137 patients with EOC and 23 non-cancer controls (Appendix A). High COL11A1 mRNA levels were significantly related to low miR-335-expressing tumors (*p* < 0.001).

Table 1 outlines the correlations between miR-335 mRNA levels and patient demographics. When compared with high miR-335 mRNA levels, low levels were associated with advanced-stage cancer (*p* = 0.005), poor response to chemotherapy (*p* = 0.001), short PFI (*p* = 0.010), and death (*p* < 0.001). Despite no significant disparity between serous and non-serous histology, miR-335 levels displayed an increasing trend in mucinous and clear cell carcinoma tissues, compared to those in endometrioid and serous histology (*p* = 0.057).

Moreover, patients with low miR-335 mRNA levels had significantly shorter OS and PFS (median OS, 34.4 months vs. not reached, *p* < 0.001; median PFS, 8.7 months vs. not reached, *p* < 0.001, respectively) than those with high miR-335 mRNA levels (Figure 4A). In the non-serous subgroup (*n* = 61), patients with low miR-335 mRNA levels also displayed significantly unfavorable prognosis (median OS, 24.6 months vs. not reached, *p* < 0.001; median PFS, 11.0 months vs. not reached, *p* < 0.001) than those with high miR-335 mRNA levels (Figure 4B). We obtained similar results in the serous subgroup (*n* = 76) (median OS, 46.2 months vs. not reached, *p* = 0.005; median PFS, 7.8 months vs. not reached, *p* = 0.006; Figure 4B).

## 4. Discussion

High COL11A1 levels are associated with tumor aggressiveness, chemoresistance, and poor survival in various solid tumor types [14,15]. The overexpression of COL11A1 has also been found in radioresistant ovarian cancer samples [32]. Recently, COL11A1 has been proposed as a therapeutic target in cancer [14]. MicroRNAs are small, non-coding RNA molecules with diverse biological functions. In cancer, the loss of tumor-suppressive miRNAs enhances the expression of target oncogenes, whereas an increased expression of oncogenic miRNAs (known as oncomirs) can repress the target tumor suppressor genes. Some miRNA and long non-coding RNA (lncRNA) were involved in the regulation of COL11A1 in cancer cells [25,26,27,33]. miR-139-5p overexpression or COL11A1 silencing could inhibit the proliferation of breast cancer cells and promote apoptosis [25]. The simultaneous overexpression of miR-139-5p and COL11A1 could reverse this effect, thereby indicating *COL11A1* as a downstream gene of miR-139-5p [25]. Another study mentioned that miRNA let-7b is a tumor suppressor and let-7b can inhibit COL11A1 expression, thereby reducing the proliferation, invasion, and migration of breast cancer cells [26]. The lncRNA small nucleolar RNA host gene 12 (SNHG12) is overexpressed in various cancer types. Xu et al. reported the involvement of miR-200c-5p in the regulation of COL11A1 by SNHG12 in renal cell carcinoma [31]. Moreover, miR-29 is involved in colon cancer progression by targeting COL11A1 [27]. Taken together, the miRNAs regulation of COL11A1 can be specific to cancer types. The association of miRNAs and lncRNA with COL11A1 in ovarian cancer still warrants investigation.

In this study, we investigated the role of miRNAs in regulating COL11A1 expression in EOC. First, we identified miR-509 and miR-335 as candidate miRNAs through an online database search. The manipulation of miR-335 RNA level, but not miR-509, regulated the mRNA expression of COL11A1. The binding and inhibition of miR-335 on COL11A1 3′UTR in A2780CP70 cells were confirmed by the luciferase reporter gene assay. The binding of miR-335 to COL11A1 could inhibit COL11A1 transcription and downregulate COL11A1 expression, thereby reducing the invasiveness and chemoresistance of EOC cells via the Ets-1/MMP3 and Akt/c/EBPβ/PDK1 axis, respectively. These in vitro results were reinforced by in vivo findings that low miR-335 levels were associated with advanced-stage cancer, poor response to chemotherapy, and poor survival. Furthermore, we observed the association between miR-335 expression and survival in both serous and non-serous EOC subgroups.

Researchers have identified the aberrant downregulation of miR-335 in breast cancer cell lines and tissues [34], gastric cancer cell lines [35], clear cell renal cell carcinoma tissues [36], and prostate cancer cell lines [37]. On the contrary, the elevated miR-335 expression has been demonstrated in colorectal cancer tissues [38], myeloma cell lines [39], and glioma cell lines [40] and tissues. Therefore, the role of miR-335 depends on the cancer type. Previous studies have associated miR-335 with cell migration regulation [41], invasiveness [42], and drug resistance [43] in EOC cells, besides identifying it as an independent prognostic factor of survival and recurrence [44]. Nevertheless, its precise mechanism of action and the regulation of COL11A1 remain unexplored in ovarian tumors. We identified COL11A1 using an online prediction algorithm to search for miR-335 control targets involved in ovarian cancer proceeding, which supposedly has a miR-335 combining site in the 3′UTR.

miR-335 mimics can suppress p-Akt, COL11A1, and PDK1 expressions, to overcome drug resistance in EOC cells. Chemoresistance still remains an important limitation to successfully treat ovarian cancer. To overcome the resistance to chemotherapy, researchers have developed agents targeting the PI3K/AKT/mTOR axis but with limited single-agent activity. mTOR inhibition can induce co-stimulatory loops that activate PI3K and AKT and are responsible for drug resistance [45]. Another possible mechanism of EOC chemoresistance may be related to PDK1, which is involved in tumor growth and progression, cancer cell invasion and dissemination, and chemoresistance [46,47]. Akt inhibition induces feedback loops that promote drug resistance by COL11A1 via increased binding activity between PDK1 and COL11A1 [20]. Furthermore, we investigated Akt inhibitors that suppress Akt and its downstream effectors 4E-BP1 and p70S6 kinase. However, most Akt inhibitors cannot inhibit COL11A1 and PDK1, and PDK1 could phosphorylate Akt, even after an Akt inhibitor-mediated decrease of COL11A1 [48]. The findings that miR-335 suppressed PDK1 expression through COL11A1 inhibition suggested that miR-335 may have potential use in patients with COL11A1-positive EOC.

## 5. Conclusions

This is the first study to demonstrate miR-335 expression and its association with COL11A1 in various ovarian cancer cell lines and clinical specimens. We analyzed miR-335 expression in the TCGA EOC database, which consists of 483 high-grade serous EOC samples (Appendix A) and compared it with our findings. The miR-335 expression in the TCGA cohort was downregulated in the advanced stage (Appendix A) and unrelated to the outcome of debulking surgery (Appendix A) and OS (Appendix A). Discrepancies between the TCGA database and ours might be attributed to differences in the sample size and impact of non-serous histological subtypes in our cohort. Despite the OS disparity in the high-grade serous subtype, our results extend the understanding of the role of miR-335 in EOC by highlighting the importance of miR-335 in both serous and non-serous subtypes of ovarian cancer.

## Figures and Tables

**Figure 1 cancers-13-06257-f001:**
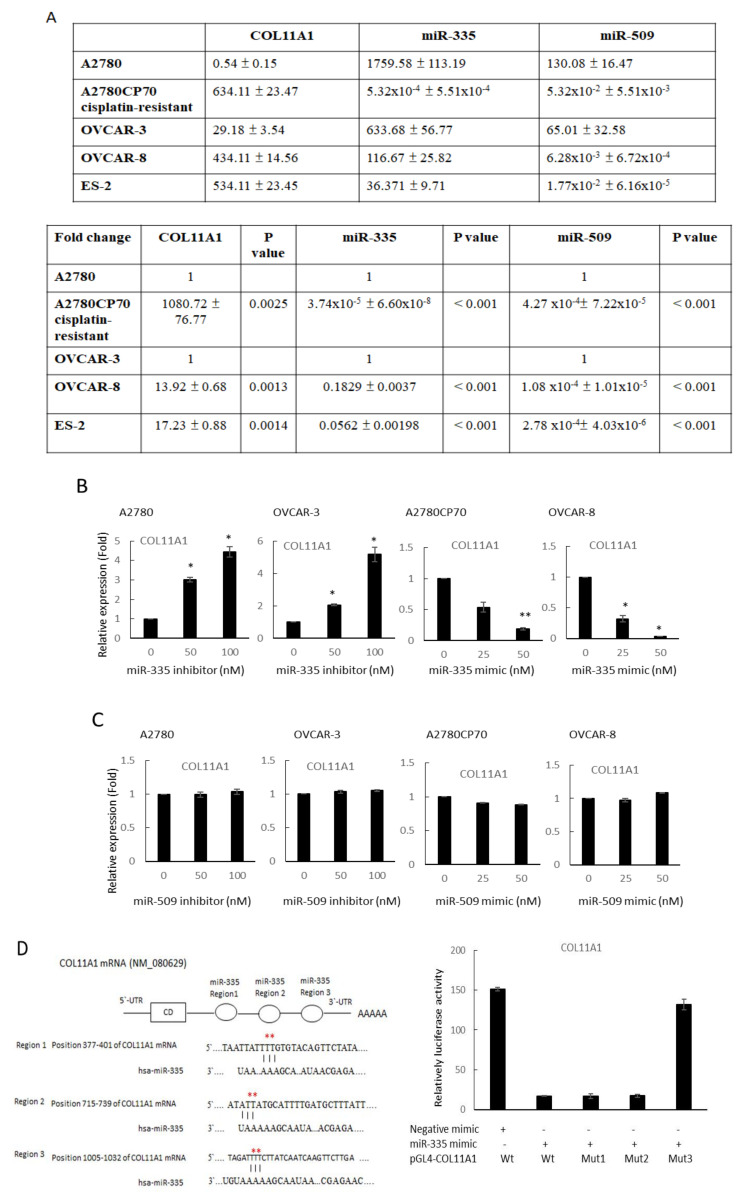
miR-335/miR-509 expression in epithelial ovarian carcinoma cells and their correlation with collagen type XI alpha 1 expression. (**A**) Messenger RNA (mRNA) expression levels of collagen type XI alpha 1 (COL11A1), miR-335, and miR-509-3p, in a panel of five ovarian cancer cell lines, were evaluated by real-time reverse transcription-polymerase chain reaction (RT-PCR). All experiments were performed in triplicate. (**B**) mRNA expression levels of COL11A1 in A2780 and OVCAR-3 cells transfected with miR-335 inhibitor and A2780CP70, as well as OVCAR-8 cells transfected with miR-335 mimic, are evaluated by real-time RT-PCR. All experiments are performed in triplicate; * *p* < 0.05, ** *p* < 0.01, compared with control. (**C**) mRNA expression levels of COL11A1 in A2780 and OVCAR-3 cells transfected with miR-509-3p inhibitor, as well as A2780CP70 and OVCAR-8 cells transfected with miR-509-3p mimic, were evaluated by real-time RT-PCR. All experiments are performed in triplicate. (**D**) Left panel: the putative miR-335 binding site in its 3′UTR (region 1, 2, and 3) contains mutant wt-COL11A1 and corresponding mut-COL11A1 (red star). Right panel: wt-COL11A1/mut-COL11A1 vector and miR-335 NC/miR-335 mimic were co-transfected into A2780CP70 cells. Compared with the control group, miR-335 mimic transfection significantly reduced the luciferase activity of the wt-COL11A1 reporter gene. In co-transfected cells with miR-335 and mut-COL11A1 (region 3) reporter genes, there was no significant decrease in reporter gene activity.

**Figure 2 cancers-13-06257-f002:**
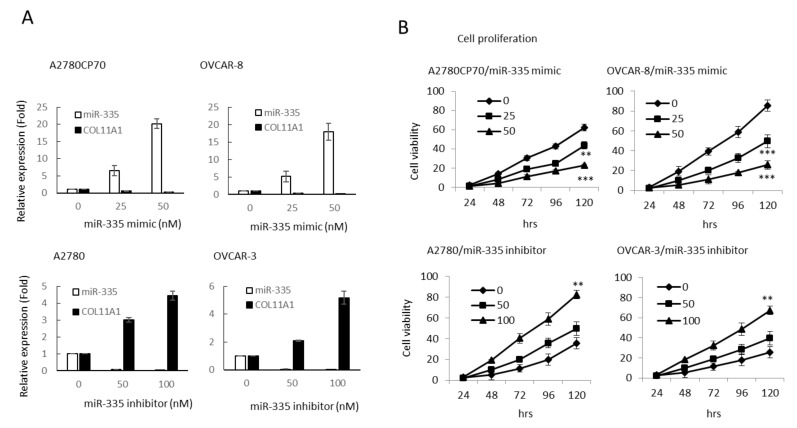
Invasive phenotypes of epithelial ovarian carcinoma cells are regulated by miR-335. (**A**) Messenger RNA (mRNA) expression levels of collagen type XI alpha 1 (COL11A1) and miR-335 in A2780CP70 and OVCAR-8 cells transfected with miR-335 mimic, as well as A2780 and OVCAR-3 cells transfected with miR-335 inhibitor, were evaluated by real-time reverse transcription-polymerase chain reaction (RT-PCR). All experiments were performed in triplicate. (**B**) The cell growth of A2780CP70 and OVCAR-8 cells transfected with miR-335 mimic and A2780 and OVCAR-3 cells transfected with miR-335 inhibitor for 48 h. Cell viability was assessed by 4,5-dimethylthiazol-2-yl)-2,5-diphenyltetrazolium bromide assays. All experiments were performed in triplicate; ** *p* < 0.01, *** *p* < 0.001, compared with control. (**C**) pCMV6-AC-GFP-COL11A1 and miR-335/NC were co-transfected into A2780CP70 and OVCAR-8 cell lines. The experimental results of MTT displayed that the cell proliferation rate was obviously decreased after transfection with the mimic of miR-335, while the overexpression of COL11A1 turned over the apparent inhibition of cell proliferation by miR-335. NC: negative control; ** *p* < 0.01, *** *p* < 0.001. (**D**) The invasive ability of A2780CP70 and OVCAR-8 cells transfected with miR-335 mimic and A2780, as well as OVCAR-3 cells transfected with miR-335 inhibitor for 48 h. All data represent the mean ± SD of three separate experiments; ** *p* < 0.01, compared with control. (**E**) Upper panel: the protein expression of COL11A1, Ets-1, and MMP3 in A2780 cells transfected with miR-335 inhibitor, as well as A2780CP70 cells transfected with miR-335 mimic, were evaluated by western blot. β-Actin is used as a protein loading control. Lower panel: MMP3 activity is evaluated in A2780 cells transfected with miR-335 inhibitor, as well as A2780CP70 cells transfected with miR-335 mimic (Appendix A). All experiments were performed in triplicate.

**Figure 3 cancers-13-06257-f003:**
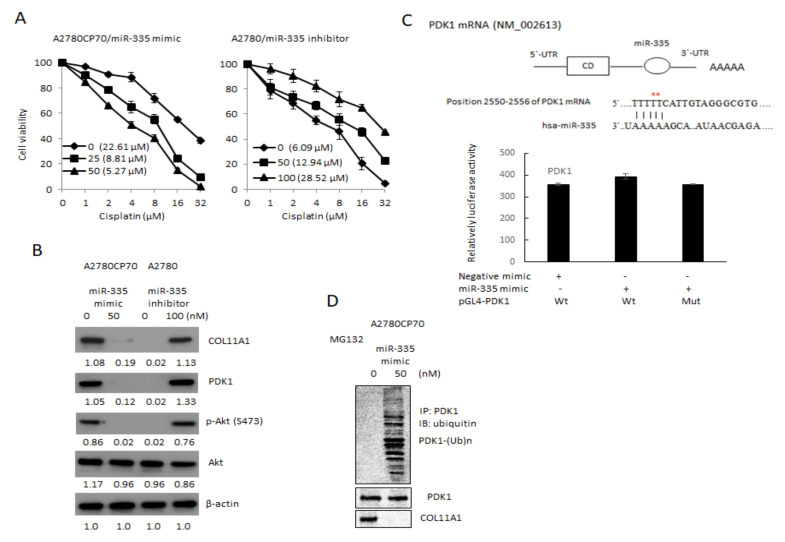
Drug sensitivity of epithelial ovarian carcinoma cells is regulated by miR-335. (**A**) A2780CP70 cells transfected with miR-335 mimic and A2780 cells transfected with miR-335 inhibitor for 48 h, subsequently treated with different concentrations of cisplatin for 48 h. Cell viability was assessed by 4,5-dimethylthiazol-2-yl)-2,5-diphenyltetrazolium bromide assays. All experiments were performed in triplicate. (**B**) The protein expression of collagen type XI alpha 1, PDK1, p-Akt (S473), and Akt in A2780 cells transfected with miR-335 inhibitor, as well as A2780CP70 cells transfected with miR-335 mimic, were evaluated by western blot. β-Actin is used as a protein loading control (Appendix A). (**C**) Upper panel: the putative miR-335 binding site in its 3′UTR contains mutant wt-PDK1 and corresponding mut-PDK1 (red star). Lower panel: wt-PDK1/mut-PDK1 vector and miR-335 NC/miR-335 mimic were co-transfected into A2780CP70 cells. Compared with the control group, miR-335 mimic transfection did not significantly decrease the luciferase activity of the wt-PDK1 reporter gene. In co-transfected cells with miR-335 and mut-PDK1 reporter genes, there was no significant decrease in reporter gene activity. (**D**) A2780CP70 cells transfected with miR-335 mimic were treated with MG132 for 6 h, and the cell lysates were immunoprecipitated with anti-PDK1 antibodies. The resulting IPs were analyzed by IB, using an anti-ubiquitin antibody (Appendix A).

**Figure 4 cancers-13-06257-f004:**
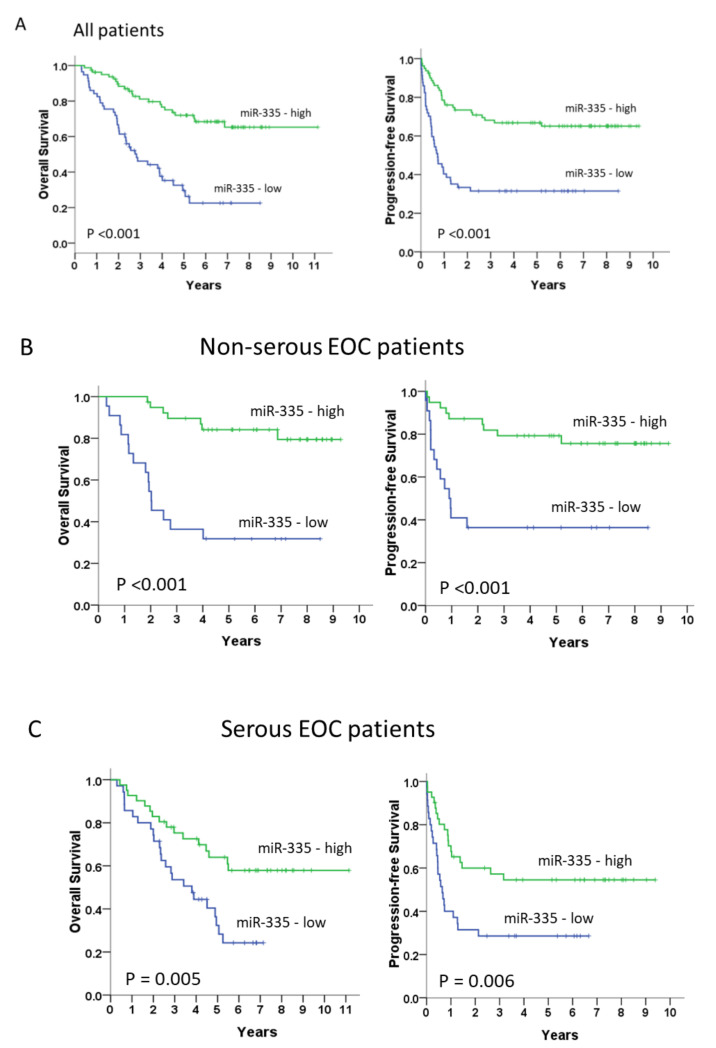
Long-term overall and progression-free survivals (**A**) in all patients (*n* = 137) and the non-serous (*n* = 61) (**B**) and serous subgroups (*n* =76) (**C**). Kaplan–Meier curves, stratified by the miR-335 messenger RNA level and assessed by log-rank test.

**Table 1 cancers-13-06257-t001:** Correlations between miR-335 mRNA levels and patient demographic (*n* = 137).

Characteristics	*N*	miR-335 mRNA Level		*p*
		Low	High	
Age (years)				0.070
<52	63	21 (33.3)	42 (66.7)	
≥52	74	36 (48.6)	38 (51.4)	
Stage				0.005
I–II	65	19 (29.2)	46 (70.8)	
III–IV	72	38 (52.8)	34 (47.2)	
Histology				0.239
Serous	76	35 (46.1)	41 (53.9)	
Non-serous	61	22 (36.1)	39 (63.9)	
Mucinous	11	1 (9.1)	10 (90.9)	
Endometrioid	11	6 (54.5)	5 (45.5)	
Clear cell	39	15 (38.5)	24 (61.5)	
Chemotherapy	109			0.001
CR/PR	79	29 (36.7)	50 (63.3)	
SD/PD	30	22 (73.3)	8 (26.7)	
PFI	109			0.010
≥6 months	79	31 (39.2)	48 (60.8)	
<6 months	30	20 (66.7)	10 (33.3)	
Death				<0.001
No	75	18 (24.0)	57 (76.0)	
Yes	62	39 (62.9)	23 (37.1)	

Data were presented as frequency (percentage). Data were analyzed by the Chi-square test or Fisher’s exact test; miR335 level <16.95 (low) and ≥16.95 (high). Abbreviation: *FIGO*, the International Federation of Gynecology and Obstetrics; CR, complete response; PR, partial response; SD, stable disease; PD, progressive disease.

## Data Availability

No new data were created or analyzed in this study. Data sharing is not applicable to this article.

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
