# Peer review of "miR-335 Restrains the Aggressive Phenotypes of Ovarian Cancer Cells by Inhibiting COL11A1"

_cancers, 2021, doi:10.3390/cancers13246257_

Round 1
Reviewer 1 Report
Add a paragraph at the end of intro to summarize the findings of the study
The methods section discusses pCMV6-AC-GFP vector carrying COL11A1. Where and why was this used?
Was mir-355 also identified on miRWalk? What is the difference between miRWalk and miRTar?
Describe the algorithms in methods section
The first sentence of results talks about overexpressing miR335? Where is the data for that?
Fig 1A. represents mean fold change? Representative experiment? Show statistics
What does it mean that COL11A1 levels didn’t change with changes in miR509?
Are there any other targets to mir335 that drive proliferation of cancer cells?
Rescue experiments should be designed to determine if addition of mir mimetic to the inhibitor group would increase the expression of COL11A1
How do you know that the effect of mir inhibitor or mimetic on cell proliferation is dependent on COL11A1? In COL11A1 knockout background will the miR inhibitor promote cell proliferation?
Does Col11A1 knockdown phenocopy mir activation in cisR?
Is there a correlation between mir335 expression and overall survival in publicly available ovarian cancer datasets?
Discuss the therapeutic implications of these findings. Given that COL11A1 expression is elevated in recurrent cancers, is there any data to show the expression of miR335 in recurrent EOC?
The manuscript needs thorough English language editing. It clearly looks like the patient data sections are well written than the rest of the manuscript
Author Response
Reviewer 1
- Add a paragraph at the end of intro to summarize the findings of the study
Response: Thanks for your suggestions. The sentences have been added to the introduction section (page 2, lines 81-85).
- The methods section discusses pCMV6-AC-GFP vector carrying COL11A1. Where and why was this used?
Response: The pCMV6-AC-GFP vector carrying COL11A1 combination with miR-335 mimics was transfected into A2780CP70 or OVCAR-8 cells to verify whether miR-335 regulated the proliferation of ovarian cancer cells through COL11A1. These results have been added to the results section (page 8, lines 299-304, Figure 2C).
- Was mir-355 also identified on miRWalk? What is the difference between miRWalk and miRTar?
Response: miR-335 was not identified in miRWalk. miRTar (Chou and et al., 2017) and miRWalk (Sticht and et al., 2018) are two similar databases. Both provide a comprehensive overview of miRNA–target interactions (MTIs), including target mining of miRNAs, genes, and pathways. However, miRwalk keeps updating its prediction method, dataset, and website and provides up-to-date MTIs information for researchers.
Chou, C., Shrestha, S., Yang, C., Chang, N., Lin, Y., Liao, K., Huang, H. miRTarBase update 2018: a resource for experimentally validated microRNA-target interactions. Nucleic Acids Research 2017, 46(D1), D296-D302.
Sticht C, De La Torre C, Parveen A, Gretz N.: miRWalk: An online resource for prediction of microRNA binding sites. PLoS One 2018, 13(10).
- Describe the algorithms in methods section.
Response: The algorithms of Quantitative reverse transcriptase PCR, Luciferase reporter analysis, Transwell invasion assay have been added in the methods section (page 3, line 141; page 4, line 170; page 4, line 183).
- Fig 1A. represents mean fold change? Representative experiment? Show statistics
Response: Figure 1A has been amended according to your suggestion.
- What does it mean that COL11A1 levels didn’t change with changes in miR509?
Response: It means that COL11A1 could not be regulated by miR-509 (page 6, lines 267-268).
- Are there any other targets to mir335 that drive proliferation of cancer cells?
Response: The literature review fails to find any other target genes of miR-335 that drive cell proliferation. Our results showed that miR-335 mimic decreased the amount of cell proliferation, and overexpression of COL11A1 inhibited the repression of miR-335 on cell growth (Figure 2C). These results suggest that COL11A1 was critical for the impacts of miR-335 in the ovarian cancer cell.
- Rescue experiments should be designed to determine if addition of mir mimetic to the inhibitor group would increase the expression of COL11A1.
Response: Thanks for your suggestions. The rescue experiments have been added to the results section (page 8, lines 299-304, Figure 2C).
- How do you know that the effect of mir inhibitor or mimetic on cell proliferation is dependent on COL11A1? In COL11A1 knockout background will the miR inhibitor promote cell proliferation?
Response: Our results showed that miR-335 mimic apparently decreased the amount of cell proliferation, but the overexpression of COL11A1 inhibited the repression of miR-335 on cell growth (Figure 2C). These results suggest that COL11A1 was critical for the impacts of miR-335 in the ovarian cancer cell.
- Is there a correlation between mir335 expression and overall survival in publicly available ovarian cancer datasets?
Response: The miR-335 expression in the TCGA database was unrelated to overall survival (Supplementary Figure 1). Discrepancies between the TCGA database and ours might be attributed to differences in the sample size and the impact of non-serous histological subtypes in our cohort (page 14, lines 460-465).
- Discuss the therapeutic implications of these findings. Given that COL11A1 expression is elevated in recurrent cancers, is there any data to show the expression of miR335 in recurrent EOC?
Response: The therapeutic implications have been addressed in the Discussion section (pages 13-14, lines 441-455): “miR335 mimics can suppress p-Akt, COL11A1, and PDK1 expressions to overcome drug resistance in EOC cells…..”.
We have not yet measured the expression of miR335 in recurrent EOC specimens. When EOC recurrence was encountered, decreased expression of miR335 was correlated with short progression-free interval (<6 months) (P = 0.010), as shown in Table 1. Also, low miR335 expression was found in patients with poor PFS, indicating the positive association between its low expression and recurrent EOC (Supplementary Figures 1A & 1B).
- The manuscript needs thorough English language editing. It clearly looks like the patient data sections are well written than the rest of the manuscript.
Response: Thanks for your suggestions. This revised manuscript has been edited by a business editor.
Reviewer 2 Report
Revision of the paper
miR-335 restrains the aggressive phenotypes of ovarian cancer 2 cells by inhibiting COL11A1 3
Yi-Hui Wu 1,2, Yu-Fang Huang 3, Tzu-Hao Chang 4, Pei-Ying Wu 5, Tsung-Ying Hsieh 6, Sheng-Yen Hsiao 7, Soon-4 Cen Huang 8,† _and Cheng-Yang Chou 9,†,_*
Submitted to Cancers
Accepted with minor revision
Minor points:
Methods
- COL11A1 was cloned into pCMV vector and checked by sequencing?
- ‘All ovarian cancer specimens and non-cancerous controls were de-identified and col-107 lected with the approval of institutional review boards.’ What does de-identified mean?
- ‘The ratio of the number of copies of the target gene mRNA to the number of copies of GAPDH was calculated as 2-Ct × K (K = 106, a constant).’ Which is the method here? What is K? How is it calculated? Please add a reference, if possible.
- To make the reading fluid, the authors could merge the par.2.6 and 2.7 as the antibodies described in the latter section are the ones used for WB. The reagent MG132 if used on the cells, could be added in the section ‘Cells and Media’
- 2.8 the details of the cloning of 3’ UTR COL11A1 in luciferase vector are missing
- 2.9: why did the authors use a fluorescence microscope to count the cresyl-violet stained cells? Didn’t they use phase-contrast images?
Results
- Fig1A: please indicate which are the resistant cell line (and eventually the drug) in the table 1A. Is OVACAR-8 cell line chemoresistant?
- Did the authors performed a power analysis to understand which is the minimal number of subjects to be involved in their analysis for each group (healthy subjects versus EOC patients)? The number of EOC patients seems to be unbalanced compared to healthy subjects.
Discussion:
- Line 368: ‘Some miRNA and long non-coding RNA 368 (lncRNA) were involved in the regulation of COL11A1 in cancer cells.’ Reference is needed.
- Line 369-371: ‘miR-139-5p over-369 expression or COL11A1 silencing could inhibit the proliferation of breast cancer cells and promote apoptosis.’ Reference is needed.
- Line 382 ‘In the current study, we investigated the role of miRNAs in regulating COL11A1 expression.’ In EOC is needed.
- Line 360-380: Recently, COL11A1 has been proposed as a therapeutic target in cancer (doi: 10.1096/fj.202100054RR.) ref 14, please re-cite it. Moreover, the overexpression of COL11A1 has been recently found also in radioresistant ovarian cancer samples (doi: 10.1080/09553002.2021.1928780.). It could also be an important paper to be discussed.
- Line 418-420: ‘The findings 418 that miR-335 suppressed PDK1 expression through COL11A1 inhibition suggested that 419 miR-335 may be used as a potential therapeutic target.’ As miR-335 could become an oncomiRs or a tumor-suppressor miRNA dependently from the tissue that is analysed (line 394-397 of the discussion), it is difficult to imagine that a single miRNA could become a therapeutic target per sè. It would be necessary to identify a mechanism for the specific tissue targeting of the molecule.
General comments:
- As emerging evidences demonstrate that microRNAs could be also secreted in biofluids of cancer patients, did the authors check the expression level of miR-335 in serum samples from patients versus normal subjects? This miRNA could become an important diagnostic and prognostic molecules.
- As miRNA could be also used as therapeutic molecules, did the authors try to perform organoid culture from the tissue samples of OEC patients and treat them with mimic miR-335 to increase its level of expression? This could be propaedeutic to the translation of miR-335 use as therapeutic agent in animal model of OEC tumor.
Author Response
Reviewer 2
Accepted with minor revision
Minor points:
Methods
1.COL11A1 was cloned into pCMV vector and checked by sequencing?
Response: COL11A1cDNA was cloned into a pCMV vector and checked by DNA sequencing (page 3, line 108).
- All ovarian cancer specimens and non-cancerous controls were de-identified and collected with the approval of institutional review boards.’ What does de-identified mean?
Response: De-identification is the process used to prevent personal identity from being revealed. Following the principle of confidentiality, patient privacy connected to all ovarian cancer specimens and non-cancerous controls were protected.
- The ratio of the number of copies of the target gene mRNA to the number of copies of GAPDH was calculated as 2-Ct × K (K = 106, a constant).’ Which is the method here? What is K? How is it calculated? Please add a reference, if possible.
Response: The described method has been amended (page 3, line 143).
- To make the reading fluid, the authors could merge the par.2.6 and 2.7 as the antibodies described in the latter section are the ones used for WB. The reagent MG132 if used on the cells, could be added in the section ‘Cells and Media’
Response: According to your suggestion, par 2.6 and 2.7 have been merged into a new par. 2.6 (page 4, lines 145-152).
- 8 the details of the cloning of 3’ UTR COL11A1 in luciferase vector are missing.
Response: The primers of COL11A1 3'-UTR and PDK1 3'-UTR have been added to the methods section (page 4, lines 155-167).
- 9: why did the authors use a fluorescence microscope to count the cresyl-violet stained cells? Didn’t they use phase-contrast images?
Response: Thanks for your correction (page 4, line 180).
Results
- Fig1A: please indicate which are the resistant cell line (and eventually the drug) in the table 1A. Is OVACAR-8 cell line chemoresistant?
Response: The resistant cell line has been indicated in Figure 1A. OVCAR-8 is not a chemoresistant cell derived from the laboratory. However, OVCAR-8 cells exhibited less sensitivity to cisplatin treatment when compared with OVCAR-3 cells.
- Did the authors performed a power analysis to understand which is the minimal number of subjects to be involved in their analysis for each group (healthy subjects versus EOC patients)? The number of EOC patients seems to be unbalanced compared to healthy subjects.
Response: The differences between miR-335 and COL11A1 in tissue samples between healthy subjects and cancerous patients are compared at the very preliminary stage. The difficulty of obtaining ovarian tissue from healthy subjects has always been anticipated and resulted in the discontinuation of collecting tissue samples in healthy subjects. Hence, this study aimed to investigate the tumor-suppressive function of miR-335 in ovarian cancer, mediated by the suppression of COL11A1 expression, thereby reducing the invasive ability and chemoresistance of EOC cells via the Ets-1/MMP3 and Akt/c/EBPβ/PDK1 axis, respectively.
Discussion:
- Line 368: ‘Some miRNA and long non-coding RNA 368 (lncRNA) were involved in the regulation of COL11A1 in cancer cells.’ Reference is needed.
Response: References have been added to the discussion section (page 13, line 405).
- Line 369-371: ‘miR-139-5p over-369 expression or COL11A1 silencing could inhibit the proliferation of breast cancer cells and promote apoptosis.’ Reference is needed.
Response: Reference has been added to the discussion section (page 13, line 406).
- Line 382 ‘In the current study, we investigated the role of miRNAs in regulating COL11A1 expression.’ In EOC is needed.
Response: The words “in EOC” have been added to the discussion section (page 13, line 418).
- Line 360-380: Recently, COL11A1 has been proposed as a therapeutic target in cancer (doi: 10.1096/fj.202100054RR.) ref 14, please re-cite it. Moreover, the overexpression of COL11A1 has been recently found also in radioresistant ovarian cancer samples (doi: 10.1080/09553002.2021.1928780.). It could also be an important paper to be discussed.
Response: Thanks for your suggestions. These references have been added to the discussion section (page 13, lines 397-400).
- Line 418-420: ‘The findings that miR-335 suppressed PDK1 expression through COL11A1 inhibition suggested that miR-335 may be used as a potential therapeutic target.’ As miR-335 could become an oncomiRs or a tumor-suppressor miRNA dependently from the tissue that is analyzed (line 394-397 of the discussion), it is difficult to imagine that a single miRNA could become a therapeutic target per sè. It would be necessary to identify a mechanism for the specific tissue targeting of the molecule.
Response: Thanks for your suggestions. We have amended our manuscript according to your suggestion (page 14, line 455).
General comments:
- As emerging evidences demonstrate that microRNAs could be also secreted in biofluids of cancer patients, did the authors check the expression level of miR-335 in serum samples from patients versus normal subjects? This miRNA could become an important diagnostic and prognostic molecules.
Response: Thanks for your suggestions. We have collected serum samples from ovarian cancer patients, but serum samples from normal subjects need to be further collected. We will process these experiments in the near future.
- As miRNA could be also used as therapeutic molecules, did the authors try to perform organoid culture from the tissue samples of OEC patients and treat them with mimic miR-335 to increase its level of expression? This could be propaedeutic to the translation of miR-335 use as therapeutic agent in animal model of OEC tumor.
Response: Thanks for your suggestions. We will incorporate these experiments into our future work.
Reviewer 3 Report
The authors Yi-Hui Wu et al. submitted a manuscript ID: cancers-1478321 with title “miR-335 restrains the aggressive phenotypes of ovarian cancer cells by inhibiting COL11A1”.
The manuscript may be useful to understand the biological role miRNA in regulation of mRNA expression coding proteins in ovarian cancer cells.
It is well known that many transcripts in differentiated cells represent mRNA encoding proteins and a similarly an high number of transcripts is represented by long non-coding RNAs (ncRNAs) and small miRNA (miRNA), whose dysregulation affects different cellular processes with particularly consequences in cancer biology.
This manuscript reports that the role of different miRNAs and in particular it identified miR-335 for its role in inhibiting COL11A1 transcription level and degradation by ubiquitination process, thus reducing the invasiveness and chemoresistance of EOC cells.
Data of this study may be useful to increase the knowledge concerning new therapeutic strategies to block high levels of collagen type XI alpha 1 (COL11A1) that are associated with tumor, and they establish the importance of miR-335 in downregulating COL11A1-mediated in ovarian tumor progression.
In conclusion, I believe that the paper is suitable for publication in cancer as it is.
Author Response
Thanks for your suggestions.
